# Limb Osseointegration—How Important Is the Role of Nutrition in the Process?

**DOI:** 10.3390/nu17040606

**Published:** 2025-02-07

**Authors:** Agnieszka Wnuk-Scardaccione, Megan Shawl Cima

**Affiliations:** 1Department of Biomechanics and Kinesiology, Institute of Physiotherapy, Faculty of Health Sciences, Jagiellonian University Medical College, 8 Skawińska Street, 31-066 Krakow, Poland; 2Johns Hopkins Physical Medicine and Rehabilitation, Johns Hopkins Medical Center, Baltimore, MD 21093, USA; mshawl1@jhmi.edu

**Keywords:** osseointegration, nutrition, bone healing, rehabilitation

## Abstract

Background and aims: Osseointegration (OI) surgery of the appendicular skeleton for repair in amputees is a treatment in which a metal implant is directly fixed to the residual bone and subsequently connected to a prosthetic limb via a transcutaneous connector through a small incision in the skin. Current treatment does not consider nutritional advice for patients undergoing the OI procedure. However, since the group of patients is very heterogeneous, the results may not be always satisfactory for patients and clinicians. Furthermore, in some individuals, incorrect nutrition and diet habits may lead to complications and rejection of the implant. Methods: We created an extensive narrative evaluation by conducting a methodical search. A comprehensive search was conducted across three major databases: PubMed, Embase, and Scopus. The search was carried out in October 2024 with no time limit specified. The approach involved using specific, pre-defined search terms, which were first applied in PubMed, followed by Embase and Scopus to ensure a broad and diverse range of articles was captured. The search process was refined by considering studies published in high-impact journals, as identified based on impact factors and subject relevance. To ensure consistency and rigor, the selection criteria were applied uniformly across all databases, and irrelevant or incomplete articles were excluded. Results: Based on the specific nature of the OI procedure, it is crucial to adapt patients’ diets and nutrition after the process. To translate the findings from the reviewed literature into practical clinical guidance for osseointegration (OI) procedures, we developed dietary recommendations for both patients and clinicians, presented as proposed dietary plans and summarized in tables. These recommendations were informed by evidence from various studies, highlighting findings that were supported by randomized controlled trials (RCTs) as well as areas where the evidence remains inconclusive or unsupported by RCTs. Major vitamins and micro- and macroelements were distinguished and presented as guidelines for clinicians. Conclusions: OI is currently the most promising therapeutic options for amputees. To promote efficient tissue healing and provide energy for rehabilitation, it is recommended to follow a healthy, well-balanced diet that contains all the essential micronutrients, macronutrients, vitamins, and minerals. We also provide suggestions for future studies.

## 1. Introduction

Limb amputation is a life-changing event that affects mobility, quality of life, and daily activities. In industrialized countries, lower-limb amputations are primarily caused by atherosclerosis and diabetes [1], while in underdeveloped countries, they are caused by traumatic injuries from industrial, traffic, or combat activities [2]. Prosthetic limbs have improved over the last few decades, with significant technological improvements; however, their application remains limited. Traditionally, a custom-designed socket is used to attach a prosthetic limb to the body. The socket must fit securely to the residual limb to enhance comfort, convey the skeleton’s stresses to the ground via the interposing soft tissues, and allow the residual limb to control the artificial limb [3]. According to some studies, one-fourth of 74 participants interviewed and one-third of 935 participants interviewed were dissatisfied with their prosthesis; they reported wounds, skin irritation, and pain, and considered themselves to have a poor or extremely low quality of life [4,5]. These issues prompted the development of innovative methods of attaching prosthetic components.

Osseointegration (OI) surgery of the appendicular skeleton for repair in amputees is a treatment in which a metal implant is directly fixed to the residual bone and subsequently connected to a prosthetic limb via a transcutaneous connector through a small incision in the skin. Per-Ingvar Brånemark discovered the strong bond between rabbit bone and titanium implants, coining the term “osseointegration” and employing titanium for human dental implants in 1965 [6]. Dental implant technology has shown successful outcomes with screw fixation devices because of the small size of the bone, the high vascularity of the jaw, significant support by the surrounding teeth minimizing torsional forces that can lead to early loosening, and dental implants experiencing mostly axial compression forces. Joint replacement has demonstrated success with press-fit implants that provide a high surface area of integration and substantial porosity and rely on maximal contact with the implant’s inherent geometric features to provide rotational stability. The ideas of osseointegration for amputees are more similar to those of arthroplasty than dentistry [7].

In addition to key factors that influence osseointegration, such as surgical technique, bone quality and quantity, postoperative inflammation or infection, smoking habits, and implant material and surface, other factors, including the host’s immunological and nutritional status, should be considered [8]. In addition to promoting a balanced diet to reach a desirable general health status, increased focus has recently been dedicated to promoting the consumption of micronutrients that could have benefits for health and resistance to disease [9].

From the point of view of this procedure, the most important thing is to take care of the elements supporting bone metabolism and the healing of the soft tissues surrounding the implant. Different micronutrients regulating bone metabolism were found to influence the skeletal system. As a result, a specific diet regimen and micronutrients may play an important role in the various stages of bone implant osseointegration. Nutritional factors have a significant impact on bone tissue repair mechanisms and bone metabolism, which are critical for achieving appropriate bone restoration and optimizing osseointegration [10]. To the best of the author’s knowledge, no study has been conducted to present the recent nutritional advice for patients undergoing the osseointegration procedure.

The aim of this article was to present a rich, meaningful, and practical review of how macro and micronutrients promote greater results for patients preparing for osseointegration surgery after limb loss. This narrative review explores the impact of micro- and macronutrients on potential increased risks of infection, wound-healing processes, and bone fusion following limb osseointegration procedures. A comprehensive literature search was conducted across three major databases: PubMed, Embase, and Scopus. The search was carried out in October 2024 with no time limit specified. Initially, the search terms used were: limb osseointegration, nutrition, macronutrients, micronutrients, diet, vitamins, alcohol, and nicotine. However, this search yielded no relevant results. A second round of searches was conducted with the following keywords: bone healing, wound healing, macronutrients, micronutrients, diet, nutrition, alcohol, nicotine, and vitamins. Articles in this review include reviews, systematic reviews, meta-analyses, randomized controlled trials, experimental studies, clinical trials, animal studies, comparative studies, cohort studies, and retrospective and prospective studies. It is important to note that we included animal studies into this review to assist in the understanding of biological mechanisms and disease progressions, as well as to determine potential safety and toxicity effects that would otherwise cause harm to humans. 

Inclusion criteria were as follows: (1) studies that explored the relationship between nutrition, wound healing, and bone fusion; and (2) articles that provided relevant, complete data on the interplay of nutrition and bone healing processes.

Exclusion criteria included: (1) Articles that did not explicitly address the relationships between nutrition and bone healing or did not provide sufficient data on these dependencies; and (2) studies lacking full-text availability or those with incomplete or inconclusive findings. After applying these criteria, the final selection was made. After a thorough review of the literature, we carefully analyzed and drew conclusions from a total of 243 articles, ensuring that the findings were robust and relevant to the focus of this study.

## 2. Macronutrients

### 2.1. Proteins

Due to the extensive nature of the OI procedure, infections and reductions in bone integrity may pose threats to the healing process [11]. Specifically, a decrease in bone density and an increase periprosthetic cortical thickness have been demonstrated in lower-limb osseointegration systems.

Proteins are composed of amino acids, which serve as structural components, biochemical catalysts, and are involved in numerous essential metabolic functions [12]. Proteins make up 30% of bone mass and are necessary for bone growth and maintenance [13], in addition to making up 50% of bone volume as collagenous and non-collagenous protein from the organic matrix [13]. Ingestion of proteins allows for the stimulation of anabolic hormones and growth factors essential for bone mass micro-architecture and wound healing. More specifically, protein contributes to the stimulation of insulin-like growth factor 1 (IGF-1), which exerts direct anabolic influences on chondrocyte proliferation, osteoblast activity, bone resorption, renal conversion of active D3 vitamin for enhanced calcium absorptions, fibroblast proliferation, angiogenesis, and epidermization [13,14,15]. Previous research has demonstrated a potential negative effect of increased protein intake due to the increased acidic load, causing a subsequent release of minerals from bone to neutralize, known as the acid–ash hypothesis [13]. However, it has also been proposed that protein taken with calcium or other dairy products may mitigate this hypothesis. Two studies in particular have demonstrated no adverse response to bone mineral density, kidney, or liver function with adequate or protein-enriched meal replacements across a population of adults aged 28–62 years old with all health backgrounds [16,17]. In fact, higher protein intake has been associated with better maintenance of bone mineral density [18], including in individuals with an osteoporotic-related hip fracture [19] and in combination with calcium and vitamin D [20,21,22].

Proteins are also essential for the tissue growth and repair needed for wound healing, especially with stoma formations and related surgical-site infections with OI [23,24]. Proteins have the greatest effect on wound healing through their role in RNA and DNA synthesis, collagen formation, immune system function, and epidermal growth [24,25]. Supplementation with protein has been shown to reduce pro-inflammatory cytokines (interleukin-6 (IL-6) and interleukin-8 (Il-8)), neutrophils, and lymphocytes that can prolong the inflammatory phase of wound healing. Conversely, supplementation has increased anti-inflammatory Il-10 and expression of mRNA for growth factors IGF-1, fibroblast growth factor 2 (FGF-2), and vascular endothelial growth factor (VEGF), allowing for accelerated wound healing through the promotion of transition from the inflammatory to proliferative phase, as well as entering the remodeling phase at a faster rate.

Adults aged 65–80 are less responsive to the anabolic effects of low protein intake compared to younger adults aged 18–37 [26]. In addition to the physiologic effects of protein mentioned above, increased protein intake can lead to increased lean body mass, which can enhance quality of life in those recovering from surgery, enhance the rehabilitation process following OI procedure, and reduce the risk of falls associated with prosthetic training [26,27,28]. To maintain protein hemostasis, it is recommended for healthy, active individuals to have 0.8 g/kg/d and up to 3.0 g/kg/day, focusing on foods high in leucine, animal proteins, and whey protein, to achieve optimal recovery [29]. Table 1 provides a summary of randomized studies on the effects of proteins on wound healing and bone growth.

### 2.2. Carbohydrates

Carbohydrates are the most abundant macromolecule and are essential to the diet to maintain homeostasis [30]. They are divided into three major subgroups: simple sugars, complex carbohydrates, and glycoconjugates (covalently attached to proteins or lipids necessary for immunity and cell membrane integrity). General deconditioning following single and multi-stage surgeries will require and increase the need for nutrients to promote tissue healing as well as sustain energy levels for rehabilitation. With inadequate carbohydrate intake or storage, the body will convert stored fat and protein into accessible energy which can compromise wound healing, muscle, and tissue recovery, and in severe cases, result in muscle wasting [31].

Carbohydrates are required for structural lubrication, transportation, immune and hormonal function, and enzymatic processes, such as providing energy for glycolysis and supporting the function of glycoproteins and glycolipids involved in cellular signaling and immune responses [31]. Upon carbohydrate metabolism via glycolysis, glucose in converted to lactate. Lactate and glucose are essential for wound healing as it provides energy to epidermal and dermal cells for energy and facilitate cell adhesion, migration, and proliferation. Carbohydrates are also a key component of glycoproteins, as a glycoconjugate, to enhance cellular structure and communication. The metabolism of glucose and bone are closely linked. A more proactive clinical examination and treatment is currently advised for diabetes-associated bone disease, which has been defined by experimental investigations that have evaluated the mechanisms of the mutual crosstalk between bone and glucose homeostasis [32,33].

A high carbohydrate diet following injury or surgery will promote a protein-sparing effect [29]. It is recommended to include 3–5 g/kg or 55–60% of total caloric intake of carbohydrates to allow for adequate energy intake and storage in a healthy adult. Such carbohydrates should include complex carbohydrates, including whole grains, fruits, vegetables, and dairy. There is limited direct evidence from randomized controlled trials (RCTs) linking carbohydrate intake specifically to tissue healing time. However, ingesting greater than 60% of carbohydrates can lead to a hyperglycemic state, which can compromise tissue healing time and immune function [34]. Table 1 provides a summary of randomized studies on the effects of carbohydrates on wound healing.

### 2.3. Lipids

The inflammatory response initiated post-operatively is necessary for appropriate tissue healing; however, it can become detrimental with prolonged inflammatory phases. Dietary fat is essential to reduce the severity and time of the inflammatory phase of tissue healing as it is an essential source of energy for wound healing, cell proliferation, and signaling molecules. Naturally occurring fats can be classified into subcategories based on their structure, including saturated fatty acids (SFA), monounsaturated fatty acids (MUFA), and polyunsaturated fatty acids (PUFA) [35]. PUFAs can be further subclassified into omega-3 and omega-6 PUFAs, which have conflicting roles in the inflammatory process [35,36].

Saturated fatty acids are necessary for fuel; however, they are typically found in animal products and processed foods, so should be included based on recommended dietary standards [35]. Diets high in Omega-6 PUFA are correlated with inflammation, vasoconstriction, and blood clotting, which are necessary to protect against infection, yet when ingested in excess, can result in elongation of the inflammatory response [36,37]. Such diets include vegetable oils, processed meat, and fried foods. When consumed in excess, there can be negative effects on bone health, reductions in bone mineral density through decreased cancellous bone content, reductions in bone formation biomarkers, and increased osteoclast-specific genes [37].

Omega-3 PUFAs produce an anti-inflammatory response via reductions in serum levels of PGE-2 (prostaglandin E2) and Il-1 (interleukin–1), as well as promotion of bone metabolism [38]. Omega-3 PUFAs can be found in dark-flesh fish, including mackerel, swordfish, sardines, and salmon [39]. Fatty acids can also be used topically in the form of virgin coconut oil, demonstrating a reduction in inflammatory phase time, a reduction in time to epithelialization, increased total levels of collagen and elastin formation, and increased fibroblast proliferation [40].

Dietary guidelines recommend that healthy adults consume 20–25% of daily caloric intake from fats, or 0.8–2 g/kg/day. Within those guidelines, 2 g/day should be from omega-3 sources, and 10 g/day from omega-6, with emphasis on the quality of the type of fat in the diet [34]. No adequate and potent experimental designs have been used in human studies to conclusively determine how ketogenic diet therapy affects bone health. Table 1 provides a summary of randomized studies on the effects of lipids on wound healing and bone growth.

**Table 1 nutrients-17-00606-t001:** Randomized controlled trials concerning the effect of macronutrients on bone and wound status.

Macronutrient	Author/Year	Animal/Human Study	Interventions	Bone Status	Wound Status
Protein	Wang X./2022 [14]	Animal (40 rats)	Group 1 (control): 8 rats, 8.3750 g/kg/day saline solution; Group 2 (model): 8 rats, 8.3750 g/kg/day saline solution; Group 3 (trail): 8 rats, whey protein group: 8.3750 g/kg/day whey protein; Group 4 (trail, low-dose compound protein): 8 rats, 4.1875 g/kg/day compound protein; and Group 5 (high-dose compound protein): 8 rats, 8.3750 g/kg/day compound protein	Not tested	Protein-treated mice showed decreased interleukin (IL)-6, IL-8, neutrophils, and lymphocytes and increased IL-10, albumin, prealbumin, total protein levels, insulin-like growth factor 1 (IGF-1), fibroblast growth factor 2 (FGF-2), and vascular endothelial growth factor (VEGF) expressions.
Protein	Schurch MA./1998 [15]	Humans (82 patients; mean age, 80.7 +/− 7.4 years)	Group 1: 42 patients, protein supplementation, 20 g/d, Group 2: 42 patients, isocaloric placebo (among controls) 6-month intervention period	Decrease in proximal femur BDM (−2.29% +/− 0.75% and −4.71% +/− 0.77% at 12 months; difference, 2.42 percentage points [CI, 0.26 to 4.59 percentage points]; *p* = 0.029).	Not tested
Protein	Tidermark J./2004 [19]	Humans (60 women;	Group 1 (control): 20 women, standard diet Group 2: 20 women on protein-rich liquid formula alone Fortimel, 200 mL/day, 20 g protein/day) Group 3: 20 women on formula combination with nandrolone decanoate 6-month intervention period	LBM decreased in the C (−1.2 +/− 2 kg) and PR groups (−1.2 +/− 1 kg)	Not tested
Carbohydrates	Carter JD/2006 [33]	Humans (30 patients)	Group 1: 15 patients consumed less than 20 g of carbohydrates per day for the 1st month and then less than 40 g per day for months 2 and 3 Group 2 (control): 15 patients with no restrictions on diet	The diet did not increase bone turnover markers compared with controls at any time point	Not tested
Carbohydrates	Tang W/2024 [34]	Humans (92 patients scheduled for daytime oral surgery)	Group 1 (control): 45 patients on midnight fasting Group 2: 47 patients in the carbohydrate–Outfast loading group (patients in the 2nd group also fasted but received the Outfast drink (4 mL/kg) 2–3 h before the induction of anesthesia) Results assessed 24 hours after administration	Not tested	Seven parameters representing patient well-being were evaluated (thirst, hunger, mouth dryness, nausea and vomiting, fatigue, anxiety, and sleep quality) on a numeric rating scale (NRS, 0–10) were lower in the 2nd group than in the 1st group postoperatively.
Lipids	So J/2020 [36]	Humans (21 patients)	Group 1: supplementation with 3 g/day EPA (eicosapentaenoic acid) Group 2: 3 g/day DHA (docosahexaenoic acid) in a random order; two phases of 10-week supplementation separated by a 10-week washout	Not tested	EPA and DHA had distinct effects on monocyte inflammatory response, with a broader effect of DHA in attenuating pro-inflammatory cytokines.
Lipids	Nevin KG/2010 [40]	Animal (rats)	Group 1—control; Group 2—treated with 0.5 mL VCO (virgin coconut oil); Group 3—treated with 1 mL VCO Treatment was administered for 10 days, and effects were monitored for an additional 14 days following treatment	Not tested	The granulation tissue weights of the treated animals (132.7 mg for group 2, and 157.7 mg for group 2) were significantly changed compared to the control group (59.0 mg).

## 3. Micronutrients

### 3.1. Calcium

In the case of rehabilitation after osseointegration surgery, taking care of the quality and structure of the bones is extremely important. The primary component of bone, calcium (Ca), can be obtained through the diet in sufficient amounts to greatly slow down the loss of bone. This micronutrient is crucial for bone growth, maintenance, and development, as well as the stability of the cellular cytoskeleton. It affects numerous extracellular and intracellular processes [41]. An adult human body contains approximately 1200 g of calcium overall, or 2% of body weight [42]. Numerous factors, including sex, age, food, physical activity, smoking, ethnicity, genetics, and endocrine disorders, influence the amount of calcium in bones [43]. To preserve calcium homeostasis when intake is decreased, the body needs to accelerate the osteolysis process. Taking care of the appropriate levels of this micronutrient is extremely important in the case of elderly patients and women in the menopausal period undergoing osseointegration treatment [44].

It is important to also mention calcium loss associated with reduced activity, as this is a problem for most patients after limb amputation. Loss of proteins from the bone matrix, which is linked to the loss of proteins from other bodily components, particularly the muscles, is the primary reason for the Ca loss from the bones during periods of inactivity [45]. Increased dietary calcium consumption has been shown to quickly increase bone mineral density (BMD) [46].

Calcium is found in dairy and fortified foods (such as orange juice, tofu, and soy milk) and is roughly twice as abundant in certain green vegetables (bok choy, broccoli, and kale). Calcium absorption is often boosted when calcium is well solubilized and inhibited when substances that bind calcium or create insoluble calcium salts are present. Oxalic acid-rich foods include the spinach plant, collard greens, sweet potatoes, strawberries, and beans, whereas phytic acid-rich foods include fiber-rich whole-grain products like wheat bran, legumes, seeds, nuts, and soy isolates [47]. The RDA for calcium in a healthy adult is 1000 mg/d for men and 1200 mg/day for women. Table 2 provides a summary of randomized studies including the effect of calcium supplementation on bone status.

### 3.2. Magnesium

Magnesium (Mg), an element required for optimal bodily function, participates in numerous metabolic pathways in cells. It is required for calcium absorption and metabolism as well as the conversion of vitamin D into its active form [48]. According to current research, Mg shortage creates an inflammatory response that includes the activation of leukocytes and macrophages, the release of inflammatory cytokines and acute-phase proteins, and an increase in free radical generation [49]. Deficient magnesium intakes in humans are generally regarded as marginal to moderate, i.e., between 50% and 90% of the recommended daily allowance (RDA). Magnesium deficiency may also have a negative impact on bone stiffness and strength [50].

Data also indicate that a higher Mg content may have a negative impact on bone metabolism and parathyroid function, resulting in deficient mineralization. Furthermore, Mg is an opponent of Ca. It is reasonable to suppose that elevated Mg levels alter the Ca/Mg ratio, resulting in dysregulation of cell activity [51]. Magnesium also plays a crucial role in wound healing by promoting cellular repair, reducing inflammation, and supporting collagen production [52]. Despite magnesium being a common mineral, there is no major meal that contains a significant amount of magnesium. Magnesium-rich foods include unprocessed (whole) grains, spinach, almonds, legumes, and potatoes (tubers). White potatoes, oven-baked potatoes, and French fries contribute to nutrient demands in children and adolescents [53]. According to reports, these veggies provide at least 5% of the recommended magnesium consumption of 420 mg/day for males and 320 mg/day for females in a healthy state. It is worth mentioning that some research has shown that smokers have lower magnesium levels in the femoral head and the femoral spongy bone than nonsmokers [54]. Changes in Mg concentrations in the body can also be produced by alcohol and coffee consumption; nutrition; stress; and the progression of certain disorders such as heart failure, kidney disease, atherosclerosis, neoplastic diseases, hypertension, diabetes, and postmenopausal osteoporosis [55]. Table 2 provides a summary of randomized studies including the effects of magnesium on wound healing status.

### 3.3. Fluoride

Fluoride (F), according to some experts, is a micronutrient required for healthy development. However, in the case of this nutrient, determining its concentration in the human body is critical because the difference between a tolerable and a toxic dose is quite small [56]. Fluoride’s effects on bone appear to be mediated on multiple levels. Fluoride can physicochemically interact with the bone mineral matrix. Fluorides enhance osteoblast growth while inhibiting osteoclast activity, resulting in increased bone mass [57,58]. Fluoride compounds are utilized to treat osteoporosis. It has been discovered that administering low doses of F- and 1,25(OH)2 D3 with steroid therapy lowers the incidence of vertebral fractures and protects against bone loss in these patients [59]. Excess fluoride consumption results in skeletal fluorosis, a disorder identified by radiographic bone alterations ranging from osteoporosis to osteosclerosis. This situation may also disrupt bone turnover, modifying the differentiation of osteoblasts and osteoclasts and causing bone abnormalities to develop [60]. As a result, there is an imbalance between bone growth and bone resorption. In the human body, 93–97% of fluorine is deposited in hard tissues, with the remainder accumulating in organs such as the liver and kidneys. Fluoride increasingly accumulates in bone over time. F levels were higher in patients over 60 years old [61]. In humans, the dominating route of fluoride absorption is via the gastrointestinal tract. Fluoride ion is produced by fluoride compounds that are either added or naturally exist in drinking water. As a result, fluoride in drinking water is generally bioavailable. The World Health Organization (WHO) recommends that the maximum limit of fluoride in drinking water be 1.5 mg/L, as well as a recommended daily allowance of 3–4 mg/day for healthy men and women, respectively, in the diet. Because fluoride bioavailability is often lowered in humans when ingested with milk or a calcium-rich diet, residents of fluoride-contaminated locations should integrate calcium-rich items into their regular diets [62]. Table 2 provides a summary of randomized studies assessing the effects of fluoride on bone status.

### 3.4. Potassium, Sodium

Because potassium (K) interacts with other nutrients (such as sodium (Na)) and processes (such as urinary calcium excretion), the relationship between potassium consumption and bone health cannot be investigated in isolation. Excess sodium consumption, as shown by salt consumption, is a known risk factor for osteoporosis. However, one study found that urine salt excretion and bone health are inversely associated [63,64]. Sodium and potassium are more concerned with neuron and muscle function than with bone repair. Maintaining overall electrolyte balance, including salt and potassium, is vital for overall health and contributes indirectly to the body’s ability to recover, particularly through bone healing. The most widely highlighted hypothesis for dietary potassium’s bone benefit is through its effect on acid–base balance, although the function of the skeleton in regulating pH is debatable [65]. Potassium is primarily found in fruits and vegetables. The potato contains the most potassium of any food. However, according to consumption habits, milk, coffee, chicken and beef dishes, orange and grapefruit juice, and potatoes are the top potassium sources for Americans. Potassium intake declined throughout the agricultural revolution as energy consumption switched from a range of plants, including potassium-rich tubers, to cereals and animal products with lower potassium concentrations, and then further declined with a transition to highly refined processed meals [66]. For the healthy male and female, it is recommended to consume 3400 mg and 2600 mg of potassium and no more than 2300 mg/day of sodium. Table 2 provides a summary of randomized studies including the effects of potassium on bone status and growth.

### 3.5. Resveratrol

Resveratrol (RSV) is a natural polyphenolic molecule (3,5,4′-trihydroxy-trans-stilbene) that protects plants from infections and other external stressors [67]. It can be found in a variety of plants, including peanuts (*Arachis hypogea*); blueberries and cranberries (*Vaccinium* spp.); Japanese knotweed (*Polygonum cuspidatum*), which is used in traditional Asian herbal medicine; and, most critically, grapevines. It is also a natural phytoalexin that plants synthesize de novo in response to fungal assault and ultraviolet (UV) irradiation [68,69]. For the average healthy adult, it is recommended to consume 500 mg per day to achieve its protective effects. Resveratrol has antibacterial properties against a remarkable variety of bacterial, viral, and fungal species. Among known growth factors, vascular endothelial growth factor (VEGF) is thought to be one of the most common, effective, and long-lasting signal-stimulating ones for angiogenesis in wounds. Several studies have shown that resveratrol promotes VEGF expression and thereby affects angiogenesis [70]. The study presented by Khanna et al. demonstrated that wound sites are rich in oxidants, which stimulate VEGF expression and, hence, aid in wound healing. According to their investigation, resveratrol therapy improved the oxidizing environment at the wound site and was also associated with increased VEGF and tenascin expression at the wound edge [71]. There are not many reports on the effect of substances on bone healing. However, among those that exist, a positive effect can be found. Resveratrol boosts cell activity, stimulates osteoblast development in connection with bone remodeling, and lowers alveolar bone loss by suppressing inflammation in disease-induced animal models [72]. More research into and improvement of resveratrol’s bone-healing effects, as well as investigations of their therapeutic relevance, is urgently required. Nonetheless, it represents an intriguing and potential innovative therapy regime which might also be utilized as a supplementary treatment to traditional treatment regimens.

### 3.6. Vitamin D

The significance of vitamin D in calcium homeostasis and subsequent bone mineralization is well understood. However, the effect of vitamin D on bone repair after a fracture is a considerably less researched topic. Animal studies have revealed promising results showing that sufficient nutrition could promote bone repair [73]. Vitamin D appears to play a role in every step of fracture healing through mobilizing calcium. However, there are inconsistent data indicating different quantities of metabolites during the healing period, and the mechanism is poorly understood [74]. Implantology pays special attention to vitamin D since it has a role in bone metabolism and immune system regulation. At the appropriate concentration, this prohormone has a positive correlation with the osseointegration process. Studies have demonstrated that vitamin D has significant potential in the processes of postoperative wound regeneration, dental implant osseointegration, and bone homeostasis around the implant [75,76]. Clinical studies indicate that taking 10 μg of vitamin D3 and 1000 mg of calcium supplements per day can reduce bone resorption and fracture rate while increasing bone density and total calcium levels, but only limited data support the concept that humans benefit from vitamin D supplementation in terms of osseointegration [77,78]. Heterogeneity and varied study designs limit us from making a definitive declaration about the effect of vitamin D on bone regeneration. While evidence supports the benefit of supplementation in deficient individuals (RDA for the healthy adult is 15 mcg or 600 IU), studies in those with sufficient vitamin D levels show less pronounced effects [79]. Table 2 provides a summary of randomized studies investigating the effects of Vitamin D on bone growth and wound healing status.

### 3.7. Vitamin K2

Vitamin K2 is a fat-soluble molecule commonly known as menaquinone (MK). Epidemiological studies indicate that a lack of vitamin K2 in the organism is associated with bone disease and mineralization difficulties. Menaquinone is known to act as a catalyst in the carboxylation of some proteins [80]. Several studies indicate that vitamin K2 insufficiency is linked to osteoporosis, pathological fractures, and vascular calcifications. Low-level vitamin K2 intake has recently been related to an increase in the risk of hip fracture in the general population, and therapy with vitamin K2 may reduce the relative risk of non-vertebral and hip fractures [81]. MK is primarily present in green, leafy vegetables, including spinach, cauliflower, and cabbage. It can also be found is found in fermented foods in Western diets, including butter, cow liver, curdled cheese, and egg yolk. Natto, a traditional Japanese soybean-based snack, is an essential source of protein in Japan [82]. Furthermore, consumption of natto, a vitamin K-rich Japanese dish, is related to considerably higher bone mineral density (BMD) in senior Japanese males, while high vitamin K2 dietary intake is associated with higher BMD in elderly adults of both genders. Conversely, inadequate dietary vitamin K2 consumption is linked to reduced BMD in women of all ages [82,83]. Dosages utilized in clinical research on the skeleton commonly correspond to 45 mg/day of vitamin K2 as MK-4 or MK-7; however, vitamin K2 is currently available in several combinations with vitamin D3 and Ca at a dosage of 45 mcg [83]. Table 2 provides a summary of randomized studies including the effects of Vitamin K2 on bone status.

### 3.8. Vitamin C

Vitamin C (ascorbic acid) is a vital antioxidant essential for the creation of collagen in bone and connective tissue, and it has been linked to enhanced collagen synthesis and subsequent tendon healing [84]. It is an extremely important component in bone metabolism because it regulates collagen hydroxylation and the expression of non-collagenic proteins such as alkaline phosphatase, osteonectin, and osteocalcin [85]. Severe vitamin C deficiency causes scurvy, a condition characterized by the weakening of collagenous structures, resulting in poor wound healing and compromised immunity [86]. Presently, it is unusual to identify this disease in the general population; nevertheless, some groups have a higher requirement for vitamin C, such as the elderly, alcoholics, smokers, and diabetics. In addition, preclinical and clinical investigations have revealed that vitamin C deficiency slows tissue repair and decreases collagen formation [87]. In the United States and Canada, a daily vitamin C consumption of 75 mg for adult women and 90 mg for adult men is suggested. In addition, pregnant women should increase their consumption by 15 mg/day, breastfeeding women by 50 mg/day, and smokers by 35 mg/day [88]. In human investigations, researchers have not seen any improvement in bone repair because of vitamin C administration [89]. However, they did note that the benefits of vitamin C intake may only be visible in vitamin C-deficient individuals. Researchers have shown that vitamin C promotes surgical wound healing following, for example, dental implant implantation, although the study did not assess the effect of vitamin C on bone regeneration using radiography or histology [90,91]. Recommending vitamin C supplementation during the bone-healing period could be a cost-effective and simple help during treatment. However, because of the high heterogeneity of recent studies, it is impossible to suggest a specific dose or route of administration of vitamin C to enhance bone healing in humans. Table 2 provides a summary of randomized studies investigating the effects of Vitamin C on bone status.

### 3.9. Vitamin A

Vitamin A is an essential vitamin that exists in several forms, including retinols, retinals, and retinoic acids. It is recommended for a healthy adult to consume 900 mcg RAE (males) and 700 mcg RAE (females) daily [92]. Dietary vitamin A is absorbed as retinol from preformed retinoids or as pro-vitamin A carotenoids, which are converted to retinol by enterocytes. Retinoids primarily act as anti-inflammatory, comedolytic, and sebolytic substances. They promote angiogenesis, dilated capillaries in the dermis, collagen synthesis, epidermal proliferation, sun-induced cellular atypia, melanin reduction, and transitory epidermal thickening [93]. Animal studies show that vitamin A may increase both collagen cross-linkage and wound breaking strength. It enhances the impact of bone morphogenetic proteins (BMPs) on osteogenic differentiation [94]. While vitamin A influences the activity of osteoblasts and osteoclasts and is involved in the maintenance of bone integrity, studies have failed to conclusively show that vitamin A intake alone can predict bone health outcomes. In some cases, excessive intake of vitamin A has been shown to negatively impact bone health by disrupting calcium metabolism and increasing the risk of bone fractures [95] However, there is no known cause-and-effect relationship between vitamin A food intake and bone health.

### 3.10. Vitamin E

Vitamin E is a lipophilic vitamin composed of two primary groups: tocopherols and tocotrienols, which are plant-based chemicals with powerful antioxidant and anti-inflammatory activities. They also serve several physiological functions in the human body [96]. Vitamin E is commonly found in seeds, vegetable oils, and nuts, and it is recommended for a healthy adult to consume 15 mg/day. Based on an initial background search on the subject, the most proposed mechanism of action for vitamin E’s influence on bone fracture healing is based on the antioxidant’s cellular-protective properties. The activation of polymorphonuclear neutrophils during the inflammatory phase of bone fracture repair, as well as the disruption of blood flow to the bone ends, both produce oxygen free radicals [97]. There have been no published human studies on the effect of vitamin E on bone fracture repair, and only one double-blind, controlled RTC has assessed the effect of mixed-tocopherol supplementation in postmenopausal women with osteopenia [98]. Based on the limited number of varied and heterogeneous animal studies examined, the effect of vitamin E on bone fracture healing was inconclusive [99]. Table 2 provides a summary of randomized studies including the effects of Vitamin E on bone status.

### 3.11. B Vitamins

The B vitamins are one type of nutrient that have been studied for their potential effects in bone health and fracture risk. B vitamins, in general, operate as cofactors for enzymes involved in the metabolic pathways that produce energy from carbs, lipids, and proteins; daily recommended allowances for individual B vitamins are seen in Table 3 below. B vitamins also help to keep the nervous system functioning properly. There are various reviews on B vitamins and bone health, with a focus on folate and cobalamin [100]. Despite inconsistent results from observational research, a few randomized clinical trials have been conducted to investigate the effects of B6, folate, and B12 supplementation in relation with changes in bone turnover biomarkers. Human studies are still needed to discover the ideal dosage of B vitamins for their usefulness in bone results, and some studies conclude that vitamin B supplementation does not significantly lower the risk of fracture or improve bone density [101]. The relationships between these B vitamins and bone outcomes are contradictory in observational research; nevertheless, a significant number of randomized clinical trials have found no protective effects of B6, folate, or B12 in bone turnover or fracture risk reduction. Future clinical trials may involve intervening with food regimens enriched with B vitamins to investigate their impact on bone mineral density, bone turnover, and/or fracture risk in elderly people, while avoiding the potential side effects of B vitamin supplementation [102]. Table 2 provides a summary of randomized studies investigating the effects of Vitamin B on bone healing.

### 3.12. Zinc

Zinc is among the most widely distributed micronutrients. It is recognized as the most crucial trace mineral for human health. Muscles contain 50% of this mineral, bone tissue 30%, and other tissues 20%. Zinc is mostly absorbed in the small intestine, with liquids having a higher efficiency (up to about 70%) than solid foods (30%) [103]. Zinc is one of the most critical nutrients for bone tissue metabolism and skeletal system function. Years of research on the role of zinc in the metabolism and formation of bone tissue have revealed that it also plays a significant role in limiting bone resorption [104]. One treatment method would be to boost zinc levels at bone regeneration sites through oral or other systemic delivery of a zinc compound. Zinc has a high oral and systemic tolerance, indicating that systemic zinc administration could be a potential therapeutic method. Furthermore, as previously noted, several investigations have shown zinc buildup in bone tissue, particularly at mineralizing tissue locations [105,106]. When dietary zinc is limited, serum zinc levels appear to be maintained at a critical level via preferential mobilization of zinc from bone [107]. Zinc plays a critical role in wound healing due to its involvement in various biological processes such as cell proliferation and can positively affect wound healing [108]. Meat, fish, and seafood are the most zinc-rich foods. Oysters have the highest zinc content per serving of any meal, while beef accounts for 20% of zinc intake in the United States due to its widespread consumption. Zinc can also be found in eggs and milk [109]. Zinc absorption from meals varies from 5% to more than 50%, depending on the number of plant-based foods (and hence phytate) in the diet. Zinc absorption is lower in mixed meals or diets that contain both animal-based and plant-based foods than in diets or meals that primarily contain animal-based foods [110]. For healthy adults, it is recommended to consume 11 mg/day for men and 8 mg/day for women. Table 2 provides a summary of randomized studies investigating the effects of zinc on bone status and wound healing.

**Table 2 nutrients-17-00606-t002:** Randomized controlled trials regarding the effects of micronutrients on bone and wound status.

Micronutrient	Author/Year	Animal/Human Study	Interventions	Bone Status	Wound Status
Calcium	Bristow SM/2014 [44]	Humans (97 postmenopausal women)	Group 1: 38 people, Ca (1 g/d) as citrate or carbonate; Group 2: 39 people, microcrystalline hydroxyapatite (MCH) preparations; Group 3 (control): 20 people, Ca-free placebo 3-month intervention period	The citrate–carbonate and MCH doses produced comparable decreases in bone resorption over 8 h and bone turnover, significatly more that the control group.	Not tested
Magnesium	Razzaghi R/2018 [52]	Humans (70 patients with diabetic foot ulcer)	Group 1: 35 patients, 250 mg/day magnesium supplements as magnesium oxide; Group 2 (control): 35 patients, placebo for 12 weeks of supplementation	Not tested	Magnesium supplementation resulted in a significant increase in serum magnesium (+0.3 ± 0.3 vs. −0.1 ± 0.2 mg/dL, *p* < 0.001) and significant reductions in ulcer length (−1.8 ± 2.0 vs. −0.9 ± 1.1 cm, *p* = 0.01), width (−1.6 ± 2.0 vs. −0.8 ± 0.9 cm, *p* = 0.02), and depth.
Fluoride	Grey A/2013 [59]	Humans (180 postmenopausal women with osteopenia)	Group 1: 45 women, 2.5 mg fluoride; Group 2: 45 women, 5 mg fluoride; Group 3: 10 mg fluoride; Group 4 (control): 45 women received placebo 1-year intervention period	Compared to placebo, none of the doses of fluoride altered BMD at any site. The bone formation marker, procollagen type I N-terminal propeptide, increased significantly in the 5 mg and 10 mg fluoride groups compared to placebo (*p =* 0.04 and 0.005, respectively)	Not tested
Potassium	Granchi/2018 [64]	Humans (40 postmenopausal women)	Group 1: 20 women treated with: K citrate (30 mEq day^−1^), calcium carbonate (500 mg day^−1^), and vitamin D (400 IU day^−1^) Group 2 (placebo): 20 women treated with calcium carbonate (500 mg day^−1^) and vitamin D (400 IU day^−1^) 3- and 6-month measurements taken over a 6-month trial period	In patients with low 24 h citrate excretion at baseline, 30% mean decreases in BAP (bone alkaline phosphatase) and CTX (carboxy-terminal telopeptide of type I collagen) were observed at 6 months. A significant reduction was also evident when low citrate (BAP: −25%; CTX: −35%) and a low pH (BAP: −25%; CTX: −30%) were found in fasting-morning urine.	Not tested
Vitamin D	Halschou-Jensen/2023 [77]	Humans (48 patients with diabetic foot ulcers)	Group 1: 24 people, high-dose vitamin D (170 μg/day) Group 2: 24 people, low-dose vitamin D (20 μg/day) for 48 weeks	Not tested	Significantly higher rate of ulcer healing in the high-dose group, with 21 of 30 (70%) healed ulcers compared to 12 of 34 (35%) in the low-dose group (*p* = 0.012).
Vitamin D	Slobogean GP/2022 [79]	Humans (102 patients with an acute tibial or femoral shaft fracture managed with a reamed, locked intramedullary nail)	Group 1: 25 patients, 150,000 IU vit. D_3_ Group 2: 24 patients, 4000 IU vit. D_3_ Group 3: 24 patients, 600 IU vit. D_3_ Group 4 (control): 27 patients, placebo 12-month treatment period	No clinically important or statistically significant differences were detected in RUST or FIX-IT scores between groups when measured at 3 months and over 12 months.	Not tested
Vitamin K2	Knapen MH/2013 [81]	Humans (325 healthy postmeno-pausal women)	Group 1: 161 women received 180 μg menaquinone-4, MK-4/day) capsule; Group 2 (control): 164 women received placebo 3-year treatment period	K_2_ did not affect the DXA-BMD, but BMC and the FNW increased relative to placebo. In the K_2_-treated group, hip bone strength remained unchanged during the 3-year intervention period, whereas in the placebo group, bone strength decreased significantly	Not tested
Vitamin C	Ekrol I/2014 [89]	Humans (336 patients with acute frature of the distal aspect of the radius)	Group 1: 169 patients received 500 mg of vitamin C Group 2 (control): 167 patients received placebo for 50 days after fracture	There were no significant differences in patient or fracture characteristics between the treatment groups. There was no significant difference in the time to fracture healing.	Not tested
Vitamin E	Vallibhakara SA/2021 [98]	Humans (52 osteopenic post-menopausal women)	Group 1: 26 women, mixed-tocopherol 400 IU/day Group 2 (control): 26 women, placebo tablet 12-week supplementation period	In the placebo group, the CTX had increased by 35.3% at 12 weeks of supplementation versus baseline (*p* < 0.001), while in the vitamin E group, there was no significant change in bone resorption markers (*p* < 0.898).	Not tested
Vitamin B	Clements M/2022 [101]	Humans (167 adults both with lower B12 status and normal level)	Group 1: 103 patients received combined B-vitamin (folic acid (200 μg), vitamin B12 (10 μg), vitamin B6 (10 mg), and riboflavin (5 mg)) Group 2: 102 patients received an active placebo (vit.D) for 2 years	In conclusion, the findings indicate that low-dose B-vitamin intervention for 2 years had no overall effect on BMD.	Not tested
Zinc	Sadighi A/2008 [106]	Humans (60 patients with traumatic bone fracture)	Group 1: 30 patients, 1 tablet of 50 mg zinc each day Group 2 (control): 30 patients, placebo tablet. 60-day treatment	The effects of zinc supplementation on serum zinc, alkaline phosphatase activity, and fracture healing of bones were assessed	Not tested
Zinc	Momen-Heravi M/2017 [108]	Humans (60 patients with diabetic foot ulcer)	Group 1: 30 patients, 50 mg elemental zinc in tablet per day Group 2 (control): 30 patients, placebo for 12 weeks.	Not tested	Compared with the placebo, zinc supplementation was associated with significant reductions in ulcer length and width.

**Table 3 nutrients-17-00606-t003:** A list of the recommended daily dietary guidelines for vitamins and minerals for patients after limb osseointegration surgery.

Vitamin/Mineral	Daily Dietary Guideline		Food Sources
	Males	Females	
Zinc [108]	11 mg	8 mg	Oysters, beef, fortified cereals
Calcium [22,46]	1000 mg	1200 mg	Plain yogurt, mozzarella, sardines
Magnesium [49,53]	420 mg	320 mg	Pumpkin seeds, chia seeds, almonds, spinach
Fluoride [56]	4 mg	3 mg	Black tea, coffee, raisins
Potassium [66]	3400 mg	2600 mg	Dried apricots, lentils, acorn squash
Sodium [66]	<2300 mg		Deli meat, pizza, soups, prepackaged meals
Resveratrol [67]	<500 mg		Peanut butter, blueberries, grape skin
Vitamin D [31]	15 mcg (600 IU)		Cod liver oil, trout, salmon, sardines
Vitamin K [82]	120 mcg	90 mcg	Natto, collards, turnip greens, kale, spinach
Vitamin C [84]	90 mg	75 mg	Red pepper, orange, kiwi, broccoli
Vitamin A [92]	900 mcg RAE	700 mcg RAE	Beef liver, sweet potato, spinach, carrots
Vitamin B6 [96]	1.7 mg	1.5 mg	Chickpeas, beef liver, tuna, salmon, potatoes
Vitamin B9 (Folate) [100]	400 mcg DFE		Beef liver, spinach, rice, asparagus
Vitamin B12 (Cobalamin) [100]	2.4 mcg		Fish, meat, poultry, eggs, clams, oysters, dairy product
Vitamin E [96]	15 mg		Sunflower seeds, almonds, sunflower oil, safflower oil, peanut butter

## 4. Alcohol and Nicotine

Nicotine use and excessive alcohol consumption are strongly discouraged following osseointegration procedures secondary to their negative health effects [111]. Alcoholism has demonstrated a correlation with low bone mass and increased fracture risk through enhanced bone resorption, reductions in bone formation, and enhanced oxidative stress, leading to the release of pro-inflammatory cytokines [112]. A rat model with osseointegration implantation into the femur and skull and treatment with a diet of 25% ethanol resulted in delayed bone formation and a reduction in the amount of bone formed compared to a control [113]. This same effect was also found with a diet as low as 5% alcohol. Additionally, increased alcohol consumption has demonstrated a negative effect on connective tissue formation, resulting in increased wound area, reductions in angiogenesis, enhanced oxidative stress, and reductions in type I collagen and fiber formation [114]. It is advised to reduce alcohol consumption to less than two servings of alcohol per day for men and one per day for women [115].

Similarly, nicotine use has been demonstrated to be a risk factor for primary and secondary osteoporosis through increased bone resorption and production of type IV (poor) collagen. Additionally, inflammation and oxidative stress as responses to smoking demonstrate negative effects on collagen metabolism and bone metabolism, promote calcitonin resistance, and reduce bone angiogenesis [116]. In a rat model, smoking was found to reduce the trabecular and cortical bone interface with the osseointegration implant, along with reductions in bone density, resulting in poor bone quality around the implant and reduced mechanical resistance [117]. Smoking and nicotine use have also been shown to reduce oxygen delivery to tissue and, thus, reduce oxygen utilization, increase thrombogenesis, cause local tissue ischemia, and result in higher rates of wound infection [118]. Smoking cessation can achieve significant changes in tissue microenvironment within 4 weeks, with reductions in inflammatory response; however, the proliferative response will remain impaired [119]. It is strongly recommended to reduce smoking and nicotine use completely prior to osseointegration procedures.

## 5. Dietary Application

Post-operative nutrition is imperative to combat surgical stress and promote healthy tissues, especially when activity is restricted and appetite might be low. The inclusion of nutrition in the rehabilitation phase of recovery will not only allow for maintenance and improvements in tissue healing, muscle strength, and energy levels, but can also promote mood and quality of adjacent joint tissue in preparation for enhanced functional mobility. A sample diet plan is listed in Table 4a,b below; however, individuals should consult a registered dietician to ensure appropriate caloric intake and address specific nutritional needs.

Occasionally, however, one might not be able to obtain an efficient amount of essential nutrients from their diet alone and could benefit from the use of dietary supplements to meet their baseline needs. Table 4 below contains a list of the recommended daily dietary guidelines for vitamins and minerals for OI patients. However, prior to taking new supplements, it is strongly recommended for each person to not only conduct their own research, but to also consult with their doctor, pharmacist, or registered dietician to make sure they are safe for consumption due to potential drug interactions or concentrations that are too high.

## 6. Conclusions

There are multiple factors that influence the healing process following osseointegration procedures. One of the greatest controllable factors is ensuring appropriate nutritional intake and practicing positive lifestyle habits. The body is not only healing and integrating its new limb, but also undergoing extensive rehabilitation in order to return to normal functional status. To promote efficient tissue healing and provide energy for rehabilitation, it is recommended to follow a healthy well-balanced diet that contains all the essential macronutrients, vitamins, and minerals that our body cannot make on its own. Due to the lack of specific research pertaining to the impact of nutrition on the recovery process of limb osseointegration, specific recommendations cannot be made at this time. As one progresses from the pre-operative phase to rehabilitation to community reintegration, the health care team should emphasize a wholistic approach to achieve an optimal healing environment within the body. Given the rarity of limb osseointegration surgery and the limited knowledge regarding the role of nutrition in this process, it is imperative that a nutritionist be involved in every case, ideally as an integral member of the osseointegration team, to ensure comprehensive care and enable the design of well-informed clinical trials.

## Figures and Tables

**Table 4 nutrients-17-00606-t004:** (**a**). A sample diet plan to assist in achieving optimal recovery following osseointegration. (**b**). A sample diet plan to assist in achieving optimal recovery following osseointegration.

(a)	
Breakfast	¾ cup Greek yogurt with ¼ cup berries and 1 tbsp chia seeds
Snack	¼ cup of almonds
Lunch	3 oz of tuna salad or turkey on whole-grain bread with 1 cup of vegetables and 2 tbsp hummus
Snack	12 oz Protein shake
Dinner	3 oz of grilled chicken with 1 cup of vegetables and side salad (1 cup) with 2 tbsp of oil-based dressing
(b)	
Breakfast	1 cup cooked oatmeal with ¼ cup blueberries
Snack	1 hard-boiled egg
Lunch	Salad (2 cups of greens of choice) with ¼ cup chickpeas, ¼ cup quinoa, 5 cherry tomatoes, ½ cup chopped cucumbers, 2 tbsp feta, and 2 tbsp balsamic vinegar
Snack	2 tbsp nut butter and 1 cup celery
Dinner	3 oz broiled salmon with ½ cup brown rice and 1 cup vegetables

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
