# Peer review of "Limb Osseointegration—How Important Is the Role of Nutrition in the Process?"

_nutrients, 2025, doi:10.3390/nu17040606_

Round 1

Reviewer 1 Report

Comments and Suggestions for Authors

This manuscript reviews the role of nutrition in bone and wound healing in limb osseointegration procedures.

  1. Please provide a detailed Methods section, including information on which databases were searched, the date(s) of the search, and the selection criteria used for including studies.
  2. Please create separate sections for the summaries of randomized controlled trials (RCTs), Table 1, and Table 2. For instance, while Table 1 is currently placed in the 2.3 LIPIDS section, it also includes information on other nutrients, such as carbohydrates.

Reviewer 2 Report

Comments and Suggestions for Authors

In the current review the authors underlined the importance of a healthy well-balanced diet that contains all the essential micronutrients, macronutrients, vitamins, and minerals in the aim to promote efficient tissue healing and provide energy for rehabilitation in the patients with osseointegration surgery. The author's underlined that no study has been conducted to present the recent nutritional advice for patients undergoing the osseointegration procedure.

Some comments/suggestions:

1. Abstract:

-line 17: please add what do you mean by “methodical search”.

-lines 17-19, you wrote: “We analyzed numerous of articles recommending nutrition in bone and wound healing and we tried to adapt it to clinical advice in OI procedure. How did you “tried to adapt it to clinical advice in OI procedure? Please add.

2. Introduction, lines 83-87: please add how many articles you consulted initially and how many remained at the end after exclusions.

3. The diet must be introduced before OI procedure, after or both before and after OI procedure? In introduction, at line 74-75 you wrote: “after limb loss preparing for osseointegration surgery”, in the abstract, line 20 you wrote “during the process” and at point 4.1. line 478 you wrote: “Post operative nutrition is imperative to combat surgical stress”. Please clarify from the beginning of the article.

4. Point 2.1. Proteins not Protein:

Lines 93-94: You wrote: “Proteins are composed of amino acids, that serve as structural support, biochemical catalysts, building blocks, enzymes, as well as many other metabolic functions”. Enzymes are biochemical catalysts. What do you mean by building blocks? Please correct/clarify.

5. Point 2.2. Carbohydrates:

Lines 141-42 Please clarify the statement “Carbohydrates are required for enzymatic processes”.

6.Point 2.3. Lipids:

Lines 183-85: Please delete: “However, it is recommended to consult your doctor prior to application of topical substances to ensure no adverse response”.

7. You didn’t write about Table 1 at points 2.1 Proteins and 2.2. Carbohydrates. The same observation regarding Table 2.

8. Table 1:

-Carbohydrates, Tang W 2024 – Please add which are the “Seven parameters representing patient well-being”.

-Proteins, Wang X./ 2022 – It is not clear what was administered to the control and what to the study group. Please reformulate.

-For the study performed on rats (Wang X./ 2022  and Nevin KG/ 2010) add please the number of rats involved in the study.  

9. Is better to present Zn before resveratrol.

10. Point 3.9. Vitamin A, line 386 – Give please more details concerning the animal studies.

11.Table 2:

- Jehle S/ 2012 – you wrote that the study was performed on healthy men and women. Why did you present this study?

- Clements M/ 2022 - you wrote that the study was performed on Humans (167 adults). Are they healthy? Please clarify the importance of the study.

12. In Tables 3a and 3b you added “A sample diet plan”. Please add what quantities of these foods should the patient consume at breakfast, snack….

13. In Table 4 you presented the recommended daily dietary guidelines for vitamins and minerals in general or which are recommended for OI patients? Please clarify.

14. Conclusion, lines 507-508, you wrote: “It is strongly recommended to consult with 507 a physician or registered dietician to facilitate an optimal”. The sentence is not complete.

15. For proteins -line 127-130, carbohydrates - line 152, lipids - line 186-88, Vitamin D and Ca – lines 327-28, vitamin K2-line 352 and vitamin C –line 366 you wrote the amounts that must be consumed daily. These values are from the Dietary guidelines for healthy people or are for OI patients? Please specify.

Are missing the daily amounts for Mg (point 3.2), F (point 3.3), K, Na (point 3.4), Zn (point 3.12), resveratrol (point 3.5), Vitamins A (point 3.9), E (point 3.10), B (point 3.11). Please add.

16. All the abbreviation must be explained in the article. Please check.

17. Typos:

Abstract, line 16: you wrote twice “may lead to”

-pg 2, line 79  - please correct:  “from HI impact journals”.

-pg 3, line 101 – you wrote: “active D3” Add please vitamin.

Author Response

Please see all our answers in the attachment below. 
